# The Effect of Vice–Virtue Bundles on Consumers’ Purchase Intentions for Vice Packaged Foods: Evidence from Randomized Experiments

**DOI:** 10.3390/foods12173270

**Published:** 2023-08-31

**Authors:** Yating Yu, Zhaoyang Sun, Chao Feng, Xiang Xiao, Yubo Hou

**Affiliations:** 1School of Management, Zhejiang University, Hangzhou 310058, China; 2School of Psychological and Cognitive Sciences, Peking University, Beijing 100871, China; zhaoysun@stu.pku.edu.cn (Z.S.); peterfunga8@foxmail.com (C.F.); 3Beijing Key Laboratory of Behavior and Mental Health, Peking University, Beijing 100871, China; 4School of Business, Guangdong University of Foreign Studies, Guangzhou 510000, China; 5Guangzhou Nanfang College, Guangzhou 510970, China; xiaox@nfu.edu.cn

**Keywords:** packaged food, vice, virtue, bundles, healthiness, regulatory focus, purchase intention

## Abstract

Packaged foods have significantly expanded their market presence, with the utilization of vice–virtue bundles gaining momentum, particularly in the realm of vice-packaged foods. Consequently, understanding how consumers respond to vice-packaged food with vice–virtue bundles (i.e., vice-packaged food combined with virtue ingredients) becomes crucial. This research investigates this issue through four experiments employing a one-way between-subjects design, incorporating distinct stimuli and measures, and involving samples from diverse sources. In Experiment 1 (*n* = 172), Experiment 2 (*n* = 169), and the follow-up experiment (*n* = 153), variance analysis, chi-square test, and mediating analysis demonstrate that consumers are more inclined to purchase vice-packaged food with vice–virtue bundles owing to the perception of it being healthier than vice packaged food with vice–virtue bundles. Furthermore, Experiment 3 (*n* = 249) employs moderated mediation analysis, uncovering that both the heightened purchase intention for vice-packaged food with vice–virtue bundles and the mediating effect of perceived healthiness are attenuated among consumers with prevention (vs. promotion) focus. Beyond contributing to theories on packaged food consumption, vice–virtue bundles, and regulatory focus theory, these findings hold practical implications for packaged food marketing, promoting rational food choices, and enhancing healthier diets.

## 1. Introduction

Individuals frequently encounter the paradox of requiring low-calorie, nutrient-rich foods for their health while simultaneously craving high-calorie, flavorful foods to satisfy hedonic desires [1]. This usually results in the predicament of deciding between options that are “healthy but not tasty” and those that are “unhealthy but tasty” [2,3,4]. For instance, individuals often find themselves in the dilemma of choosing between a burger and a salad, where opting for pleasure seems to come at the cost of sacrificing health, while selecting health appears to lead to dissatisfaction. This dilemma hurts people’s willingness to consume and their overall consumption experience. However, when burgers are presented in a bundle alongside salads for sale, it mitigates consumers’ indecisiveness and enhances their purchase intention. Similarly, when a cake is garnished with fruit, individuals tend to underestimate its calorie content, thus being more inclined to consume it [2,5]. Thus, properly assembling bundles of virtue (i.e., healthy things) and vice (i.e., unhealthy things) may mitigate the dilemma by reshaping consumers’ perceptions and promoting their intentions to consume the bundles [6,7,8]. In addition to being carried in served-on-a-plate foods, vice–virtue bundles are also prevalent in packaged foods, such as chocolate whole-meal bread, sugar-coated walnut meat, buckwheat crisps, and corn crispy chips. The first two exemplify vice–virtue bundles that incorporate vice ingredients into virtue-packaged food (i.e., virtue-packaged food with vice–virtue bundles), while the latter two represent vice–virtue bundles that add virtue ingredients to vice-packaged food (i.e., vice packaged food with vice–virtue bundles) [6].

Prior research has primarily focused on the phenomenon regarding vice–virtue bundles in served-on-a-plate foods [2,5,9], but limited attention has been given to understanding the dynamics of these bundles in packaged foods. Unlike served-on-a-plate foods, consumers can only infer the presence of vice–virtue bundles in packaged food by examining claims on the outer packaging [10,11]. Yet, packaging claims may be repelled and discredited by consumers, resulting in inconsistent consequences with expectations [12,13]. In addition, packaged food takes an increasing share of the food market. Vice–virtue bundling phenomenon is becoming increasingly prevalent among packaged foods, particularly vice-packaged foods [6]. Does vice-packaged food with vice–virtue bundles trigger a higher intention to purchase than vice-packaged food without vice–virtue bundles? What mechanism underlies this effect, and among which consumer groups does it manifest more prominently? Addressing these proposed issues is the objective of our research, as unraveling these dynamics holds significant implications for consumers, government regulators, and marketers.

Our research hypothesizes that consumers are more inclined to buy vice-packaged food with vice–virtue bundles than vice-packaged food without vice–virtue bundles. This inclination is believed to stem from the perception that the former option is healthier based on the prior findings that packaging claims often shape consumers’ perceptions and that health-related claims frequently lead to an overestimated perception of healthiness [2,5,6,14,15]. Nevertheless, the manifestation of these hypothesized effects might hinge on consumers’ established attitudes toward packaging claims, which may be influenced by their regulatory focus. Previous studies have revealed that individuals with a promotion focus tend to believe the packaging claims and focus on the positive points, while those with a prevention focus tend to concentrate on the negative points and approach packaging claims with caution [16,17]. Accordingly, we further propose that vice-packaged food with vice–virtue bundles increases the perceived healthiness and subsequently elevates the purchase intentions among promotion-focused consumers, but these effects are attenuated among prevention-focused consumers.

The current research extends the research on both packaged food consumption and mixed bundles by identifying that vice–virtue bundles increase consumers’ perceived healthiness of the vice packaged food and boost their purchase intentions. Moreover, our work enriches the application of regulatory focus theory to the field of packaged food consumption by revealing its moderating effect on the relationship between vice-packaged food with vice–virtue bundles and purchase intention. Finally, these findings and manipulation methods yield practical implications for packaged food promotion, enhancing rational food consumption and healthy diet, and cultivating a positive market environment.

## 2. Theories and Hypotheses

### 2.1. Vice and Virtue

Food choices are often classified as virtue or vice, based on a good/bad dichotomy, as noted by Rozin et al. [3]. Prior research defines virtues as choices aligned with long-term self-control goals, which may not offer immediate pleasure [4,6,18,19]. Conversely, vices are defined as choices that satisfy immediate pleasure but do not align with long-term self-control goals. Foods typically considered inherently healthy, such as vegetables, fruits, and those labeled as “high in fiber”, “fat-free”, and “low in sugar”, fall under the virtue category. Conversely, items considered inherently unhealthy, such as crisps, chocolate, and those labeled as “full-fat”, “rich”, and “creamy” are classified as vice [4]. Vice and virtue are generally seen as two opposite ends of a continuum [2,20,21].

The vice–virtue bundles are formed when vice and virtue elements are combined [6]. The bundles can be further divided into three types based on the relative proportion of vice and virtue: heavy vice and light virtue, equal amounts of vice and virtue, and light vice and heavy virtue [4,21]. For the purpose of this research, we exclusively focus on vice–virtue bundles that consist of heavy vice and light virtue elements, that is, vice food combined with virtue ingredients.

### 2.2. Vice-Packaged Food with Vice–Virtue Bundles and Purchase Intention

When consumers assess combinations of vice and virtue options, they tend to employ an averaging rather than an additive approach to calculating the calories contained in the vice and the virtue components, leading to a systematical underestimation of the combined calorie content [2,22]. Previous research on served-on-a-plate food has demonstrated that when the cake is combined with fruits, people estimate fewer calories for this cake with vice–virtue bundles than for the cake without vice–virtue bundles [5]. Moreover, even a small increase in the proportion of virtue in vice meals can lead to an increase in the perceived healthiness of the overall meal [6]. Similarly, consumers usually overestimate the healthiness of packaged food when it is labeled with a healthy claim [14,23,24]. Additionally, the claims on the packaging have been proven to be effective in conveying information [25,26], and most consumers rely on visual cues on packaging to make judgments about packaged food [27,28,29]. Therefore, when vice-packaged food is added to the virtue ingredients, resulting in vice-packaged food with vice–virtue bundles, consumers’ perceived healthiness of the food is likely to increase.

Numerous studies have consistently indicated that the perceived healthiness of a product usually positively influences consumers’ purchase intentions [30,31,32]. Vice food options typically present people with a choice dilemma weighing short-term hedonism against long-term health goals [33]. However, an increase in the perceived healthiness of vice food alleviates the perception of the detrimental effect of seeking short-term hedonic gratification on long-term goals, which in turn stimulates people to consume it to achieve enjoyment [2,14,24,34]. Accordingly, we propose the hypotheses:

**Hypothesis** **1.**
*Consumers are more willing to purchase vice-packaged food with vice–virtue bundles than vice-packaged food without vice–virtue bundles.*


**Hypothesis** **2.**
*Increased purchase intention for vice-packaged food with vice–virtue bundles (vs. vice-packaged food without vice–virtue bundles) is mediated by the heightened perceived healthiness.*


### 2.3. The Moderating Role of Regulatory Focus

The regulatory focus theory proposes distinct goal-oriented tendencies among individuals, categorized as promotion focus and prevention focus. Promotion-focused individuals prioritize achievement and progress, seek to maximize gains, and tend to employ an approach strategy; in contrast, prevention-focused individuals are more concerned with safety, responsibility, and duty, attempt to minimize losses, and typically adopt an avoidance strategy [35,36,37,38].

Previous research has implied that promotion-focused individuals pay more attention to positive messages or positive aspects of a message, whereas prevention-focused individuals focus more on negative messages or negative aspects [16,39,40]. Thus, promotion-focused people are more likely to direct their attention towards the virtues in vice–virtue bundles, whereas prevention-focused people may lean towards considering vices. In addition, promotion-focused consumers tend to trust marketing claims, whereas prevention-focused consumers are more likely to be skeptical of marketing claims [17,41,42]. This implies that the former may tend to trust vice–virtue bundles’ claims, whereas the latter may be skeptical. Thus, we suggest that compared to prevention-focused consumers, promotion-focused consumers are more attentive to virtue cues in vice-packaged food with vice–virtue bundles and more trusting of the existence of the bundles, and consequently perceive this type of food as healthier. Therefore, we propose the hypothesis:

**Hypothesis** **3.**
*Vice-packaged food with vice–virtue bundles (vs. vice-packaged food without vice–virtue bundles) enhances perceived healthiness and subsequently increases purchase intention among consumers with a promotion focus, while these proposed effects are attenuated among consumers with a prevention focus.*


## 3. Experiment 1

Experiment 1 aimed to test whether consumers’ purchase intentions for the vice packaged food with vice–virtue bundles are higher than their purchase intentions for vice packaged food without vice–virtue bundles (H1).

### 3.1. Method

#### 3.1.1. Participants

The calculated result of G*power 3.1 indicated that the analysis of variance (ANOVA) with one-way and two-level (effect size below 0.25, α = 0.05, 80% power) [43] needs at least 128 participants. Our researchers posted an enrollment notice to recruit participants on 10 April 2021 and closed recruitment by 15 April 2021. A total of 180 participants from a university completed the laboratory experiment in exchange for credit score compensation between 16 April and 18 April 2021. Moreover, all participants at the time of the examination had no psychiatric diseases or any deadly diseases, and they possessed normal vision. We excluded eight participants for failing an attention check (e.g., “Choose the fourth option for this question”), leaving 172 participants (M_age_ = 24.70, SD = 4.74; 55.23% males). This experiment utilized a single factor (vice packaged food: virtue labeled vs. unlabeled) between-subjects design, where participants were randomly assigned to the experimental (i.e., virtue labeled; *n* = 84; M_age_ = 24.85, SD = 4.19; 58.30% males) or control (i.e., unlabeled; *n* = 88; M_age_ = 24.57, SD = 5.22; 52.30% males) condition (see detailed demographics in Appendix A).

#### 3.1.2. Materials and Procedures

A research assistant who did not know the objective of this research served as the primary tester for this study. He instructed participants to select a slip at random from a set of ten, each labeled with numbers from 1 to 10. Their assignment to either the experimental or control condition depended on whether the drawn number was odd or even. Notably, participants were not informed about their condition allocation throughout the experiment. After starting the experiment, participants were first presented with an image of packaged food. Considering that vice-packaged food without vice–virtue bundles and vice-packaged food with vice–virtue bundles were prevalent in the category of carbohydrates, the vice-packaged food used for measurement was primarily drawn from the carbohydrate group. Instant noodles are noodles that have been steamed, deep-fried, and combined with seasonings and preservatives to enable immediate consumption, prolong shelf life, and offer a savory flavor. Instant noodles are high in fat, salt, and synthetic additives. Nevertheless, oats are grains with high levels of dietary fiber and protein. Participants in the experimental condition saw the instant noodles package with the “instant noodles with added oats” claim, while participants in the control condition saw the instant noodles package only with the “instant noodles” claim [21] (see Figure 1). After that, participants were asked to report their purchase intentions for the instant noodles on a 2-item scale (α = 0.94; “I plan to buy this food”, “I have a high probability of buying this food”; 1 = not at all, 7 = very much) [44]. Notably, we conducted a pretest (*n* = 42) to ascertain if participants’ perceptions were aligned with our expectations regarding the chosen stimuli. In this pretest, we provided participants with explanations of vice and virtue (i.e., “Vice” characterizes something that gratifies immediate indulgence and hedonic, yet eventually results in negative outcomes. Consequently, in the context of food, “vice” generally signifies unhealthy dietary choices. Conversely, “virtue” serves as the contrary of vice; hence, within the realm of food, it predominantly signifies healthful dietary options) and requested them to evaluate their perception of the food they were looking at (1 = I perceive that the food is vice, 7 = I perceive that the food is a virtue) [45]. The analysis of the *t*-test confirmed that participants perceived instant noodles as a vice (M = 2.95, SD = 1.17, vs. 4 [scale midpoint]; *t* (41) = 4.16, *p* < 0.001) and perceived oats as a virtue (M = 5.73, SD = 0.96, vs. 4 [scale midpoint]; *t* (41) = 11.68, *p* < 0.001). Next, we measured several control variables on a 7-point scale (1 = not at all, 7 = extremely), including perceived hunger, perceived happiness, perceived tastiness of the presented instant noodles, perceived familiarity with instant noodles in daily life, and how much they like instant noodles in daily life [11]. Finally, participants answered their age, gender, educational background, and monthly consumption.

### 3.2. Results

The results of ANOVA showed that the purchase intention in the experimental condition (M = 4.71, SD = 1.44) was higher than that in the control condition (M = 4.06, SD = 1.43; F (1, 170) = 9.01, *p* = 0.003; η^2^ = 0.05). That effect still held when perceived hunger, pleasure, tastiness, daily familiarity, and daily likeness were controlled (F (1, 165) = 8.64, *p* = 0.004; η^2^ = 0.05). Thus, H1 was supported.

In addition, a follow-up experiment was conducted from 11 August to 14 August 2023, employing real packaged foods as stimuli (see Figure 2), using actual purchasing options as the measure, and investigating among company staff (*n* = 153; M_age_ = 30.90, SD = 4.67; 58.80% males) to replicate the test of H1 (see details in Appendix A). The chi-square test showed a significant difference in purchasing behavior (purchase vs. not purchase) between the control (i.e., exposed to the instant noodles without labeled carrot) and experimental (i.e., exposed to the instant noodles with labeled carrot) conditions (χ^2^ = 9.28, *p* = 0.002). Specifically, 48.7% of participants in the control condition opted to purchase the instant noodles, while 72.7% of participants in the experimental condition opted to purchase. Thus, H1 was repeatedly supported.

## 4. Experiment 2

Experiment 2 provided initial evidence for the mediating effect of perceived healthiness (H2), that is, consumers’ purchase intentions of the vice-packaged food with vice–virtue bundles are higher than that of vice-packaged food without vice–virtue bundles because the former food is perceived to be healthier. Additionally, this experiment repeatedly tested H1 via stimuli that differed from that in Experiment 1.

### 4.1. Method

#### 4.1.1. Participants

A total of 200 participants from MRO (A data collection platform similar to Amazon M-Turk) completed an online experiment in exchange for financial compensation between 22 April and 24 April 2021. We excluded thirty-one participants for an attention check, leaving 169 participants (M_age_ = 25.32, SD = 5.06; 57.3% males). The selection and exclusion criteria of the participants were similar to those in Experiment 1. We utilized a single factor (vice packaged food: virtue labeled vs. unlabeled) between-subjects design, where participants were randomly assigned to the experimental (*n* = 86; M_age_ = 25.45, SD = 4.36; 55.80% males) or control (*n* = 83; M_age_ = 25.18, SD = 5.71; 59.00% males) condition. The data collection platform automatically generated this random sequence and refrained from disclosing participants’ allocation to either the experimental or control condition.

#### 4.1.2. Materials and Procedures

We first told participants that we were conducting market research for a new kind of crispy chips from the “KIKI” enterprise and presented participants with its image. Crispy chips are starchy products that are deep-fried and generously seasoned, resulting in snacks that are rich in both fat and salt, while corn is a grain containing high levels of minerals, vitamins, and dietary fiber. Participants in the experimental condition saw the crispy chips package with the “crispy chips with added corn” claim, while participants in the control condition saw the crispy chips package only with the “crispy chips” claim (see Figure 3). Following this, we asked them to imagine how they felt after viewing the information on the package when they saw these crispy chips in the mall (α = 0.94; “I plan to buy this food”, “I have a high probability of buying this food;” 1 = not at all, 7 = very much). Moreover, the pretest (*n* = 42; as in Experiment 1) showed that crispy chips were perceived as a vice (M = 2.60, SD = 1.06; *t* (41) = 8.58, *p* < 0.001) and corn was perceived as a virtue (M = 6.29, SD = 0.71; *t* (41) = 20.91, *p* < 0.001) by participants. Then, participants reported the perceived healthiness on a 2-item scale (α = 0.93; “I think the food is healthy”; “I think the food is in line with a healthy diet”; 1 = not at all, 7 = very much) [14]. Finally, participants rated their perceived hunger, pleasure, tastiness, daily familiarity, and daily likeness and reported the demographics information (as in Experiment 1).

### 4.2. Results

#### 4.2.1. Main Effect

The results of ANOVA revealed that participants in the experimental condition had higher purchase intentions (M = 5.01, SD = 1.27) than those in the control condition (M = 4.27, SD = 1.56; F (1, 167) = 11.35, *p* = 0.001; η^2^ = 0.06). This effect still held when perceived hunger, pleasure, tastiness, daily familiarity, and daily likeness were controlled (F (1, 162) = 4.73, *p* = 0.03; η^2^ = 0.03). Thereby, H1 was successfully replicated.

#### 4.2.2. Perceived Healthiness as a Mediator

We examined the mediating effect of perceived healthiness by conducting the mediation analysis in SPSS (Process model 4; 5000 iterations) [46]. We included the existence of bundles on the vice-packaged food as the independent variable (1 = virtue labeled, 0 = unlabeled), perceived healthiness of the seen packaged food as the mediator, purchase intention as the dependent variable, and perceived hunger, pleasure, tastiness, daily familiarity, and daily likeness as covariate variables. The results indicated that participants reported a higher level of perceived healthiness in the experimental condition than those in the control condition (β = 0.60, *t* = 3.15; *p* = 0.002; 95% CI [0.2241, 0.9781]), and the perceived healthiness increased the purchase intention (β = 0.27, *t* = 3.73; *p* < 0.001; 95% CI [0.1256, 0.4085]). Moreover, the perceived healthiness mediated the positive relationship between the existence of bundles on the vice base packaged food and purchase intention (indirect effect = 0.16, 95% CI [0.0467, 0.3020]; see Figure 4) in support of H2.

## 5. Experiment 3

Experiment 3 tested the moderating impact of regulatory focus on the relationship between the existence of vice–virtue bundles on vice-packaged food and perceived healthiness and its moderating effect on the mediating role of perceived healthiness (H3). Additionally, we provided robust evidence for Hypotheses 1 and 2 through the stimuli that differed from that in Experiments 1 and 2.

### 5.1. Method

#### 5.1.1. Participants

We posted an enrollment notice in several communities to recruit participants on 15 May 2021 and closed recruitment by 20 May 2021. A total of 260 community members completed a laboratory experiment in exchange for financial compensation between 21 May and 24 May 2021. Eleven participants were excluded due to the failed attention check, leaving 249 participants (M_age_ = 24.59, SD = 4.43; 35.3% males). The selection and exclusion criteria of the participants were similar to those in Experiment 1. We utilized a single factor (vice packaged food: virtue labeled vs. unlabeled) between-subjects design, where participants were randomly assigned to experimental (*n* = 113; M_age_ = 25.15, SD = 4.83; 35.40% males) or control (*n* = 136; M_age_ = 24.13, SD = 4.02; 35.30% males) condition.

#### 5.1.2. Materials and Procedures

The research assistant completed the identical randomized assignments as in Experiment 1. During the experiment, participants were first told that we were conducting market research for a new kind of spicy gluten from the “KIKI” enterprise and presented participants with its image. Spicy gluten is the gluten that has been infused with significant amounts of cayenne pepper, salt, seasonings, and oil, which is notable for its elevated levels of both fat and salt. However, multigrain is considered to have the benefits of dietary fiber. Participants in the experimental condition saw the spicy gluten package with the “spicy gluten with added multigrain” claim, while participants in the control condition saw the spicy gluten package only with the “spicy gluten” claim (see Figure 5). We then measured the purchase intention by asking participants to imagine how they felt after viewing the information on the package when they saw this spicy gluten in the mall (α = 0.94; as in Experiment 2). Additionally, the pretest (*n* = 42; as in Experiment 1) showed that spicy gluten was perceived as a vice (M = 2.14, SD = 0.95; *t* (41) = 12.65, *p* < 0.001) and multigrain was perceived as a virtue (M = 6.36, SD = 0.76; *t* (41) = 20.12, *p* < 0.001) by participants, which was consistent with our expectations. Following this, participants reported their perceived healthiness (α = 0.93, as in Experiment 2). Next, participants rated their regulatory focus on a 10-item and 7-point scale (α = 0.77; 1 = not at all, 7 = very much) [47], of which six items measured the promotion focus (I feel like I have made progress toward being successful in my life”, “Do you often do well at different things that you try?”, “How often have you accomplished things that got you ‘psyched‘ to work even harder?”, “Compared to most people, I typically unable to get what I want out of life”(Reverse), “I have found very few hobbies or activities in my life that capture my interest or motivate me to put effort into them”(R), “When it comes to achieving things that are important to me, I find that I do not perform as well as I ideally would like to do”(R)) and four items measured the prevention focus (“How often did you obey rules and regulations that were established by your parents?”, “Did you get on your parents’ nerves often when you were growing up?”(R), “Growing up, would you ever ‘cross the line’ by doing things that your parents would not tolerate?”(R), “Growing up, did you ever act in ways that your parents thought were objectionable?”(R)). We judged the regulatory focus according to the value (ΔX) of X2 (i.e., the average score of the prevention focus items) minus X1 (i.e., the average score of the promotion focus items). The individual’s regulatory focus is promotion focus if ΔX > 0, while it is prevention focus if ΔX < 0 [37]. Finally, participants rated their perceived hunger, pleasure, tastiness, daily familiarity, and daily likeness and reported the demographics information (as in Experiment 1).

### 5.2. Results

#### 5.2.1. Main Effect

The results of ANOVA indicated that participants in the experimental condition (M = 5.30, SD = 1.28) reported greater purchase intention than those in the control condition (M = 4.72, SD = 1.58; F (1, 247) = 9.92, *p* = 0.002; η^2^ = 0.04). That effect still held when perceived hunger, pleasure, tastiness, daily familiarity, and daily likeness were controlled (F (1, 242) = 6.87, *p* = 0.009; η^2^ = −0.03), verifying H1.

#### 5.2.2. Perceived Healthiness as a Mediator

The results of mediation analysis indicated that participants reported a higher level of perceived healthiness in the experimental condition than those in the control condition (β = 1.04, *t* = 6.46; *p* < 0.001; 95% CI [0.7200, 1.3516]), and the perceived healthiness increased the purchase intention (β = 0.11, *t* = 2.30; *p* = 0.02; 95% CI [0.0156, 0.2025]). Moreover, the perceived healthiness mediated the positive relationship between the existence of bundles on the vice-packaged food and purchase intention (indirect effect = 0.11, 95% CI [0.0156, 0.2025]), repeatedly confirming H2.

#### 5.2.3. Regulatory Focus as a Moderator

We conducted the moderated mediation analysis in SPSS (Process model 7; 5000 iterations), where the existence of bundles on the vice packaged food (1 = virtue labeled, 0 = unlabeled) was the independent variable, perceived healthiness of the seen packaged food as the mediator, purchase intention was the dependent variable, regulatory focus (0 = prevention focus 1 = promotion focus) was the moderator, and perceived hunger, pleasure, tastiness, daily familiarity, and daily likeness were covariate variables. The results showed that regulatory focus moderated the relationship between the existence of vice–virtue bundles on vice-packaged food and perceived healthiness (i.e., the first-stage effect; b = 0.88, *t* = 2.76, *p* = 0.006; 95% CI [0.2506, 1.5036]). Specifically, the first-stage effect was significantly positive among participants with a promotion focus (b = 1.44, *t* = 6.64, *p* < 0.001; 95% CI [1.0153, 1.8726]), while the first-stage effect was significantly attenuated among participants with a prevention focus (b = 0.57, *t* = 2.45, *p* = 0.02; 95% CI [0.1110, 1.0228]). Additionally, the regulatory focus moderated the mediating effect of perceived healthiness on the relationship between the existence of vice–virtue bundles on vice-packaged food and purchase intention (moderated mediation index = 0.10, 95% CI [0.0063, 0.2182]). Specifically, the mediating effect was significant among participants with a promotion focus (indirect effect = 0.16, 95% CI [0.0173, 0.3119]), whereas the mediating effect was significantly attenuated among participants with a prevention focus (indirect effect = 0.06, 95% CI [0.0044, 0.1400]; see Figure 6). Thus, H3 was supported.

## 6. General Discussion

### 6.1. Summary of Findings

Our four experiments provide full support for H1 identifying the facilitating role of the presence of vice–virtue bundles (i.e., the addition of virtue ingredients) on vice-packaged foods, H2 indicating the perceived healthiness as a mediator, and H3 highlighting the regulatory focus as a moderator.

First, the current research shows significantly higher purchase intentions for vice-packaged food with vice–virtue bundles (i.e., vice-packaged food with added virtue ingredients) than that for vice-packaged food without vice–virtue bundles, which is robust whether it is for instant noodles (Experiment 1 and follow-up experiment), crispy chips (Experiment 2), or spicy gluten (Experiment 3). This reinforces the idea that bundling vice foods with virtue foods encourages consumption [6,21], not only within the context of served-on-a-plate foods [2,5,9] but also in the realm of packaged foods. The fact that this bundling message is communicated to consumers through claims printed on the packaging also aligns with previous research findings indicating that claims on packages can impact intentions to purchase [9,11,12,15].

Second, Experiments 2 and 3 demonstrate that individuals prefer vice packaged foods enriched with virtue ingredients due to their perceived healthiness, indicating the mediating role of perceived healthiness. This aligns with prior research showing that claims of virtuous ingredients tend to create “health halos” that lead to an overestimation of health benefits [24,48]. This finding further confirms the trend toward increased willingness to purchase healthy foods [31,32,33], a trend more pronounced with the improved living standards and aftermath of the COVID-19 pandemic [15,49].

Third, Experiment 3 reveals that the intent to purchase vice-packaged food with vice–virtue bundles and the mediating impact of perceived healthiness are greater among consumers with a promotion focus than a prevention focus. This concurs with the regulatory focus theory, indicating that promotion-focused consumers tend to prioritize positive information and trust assertions, whereas prevention-focused consumers are inclined to emphasize negative information and approach marketing claims with skepticism [37,42,50].

We reinforced the generality and robustness of these findings by utilizing a diverse set of stimuli representing various vice and virtue combinations, different measures of purchase intention, and including a wide range of samples [11,51]. These measures ensured that our findings were consistent across different scenarios and demographics. Overall, our research sheds light on consumers’ responses to vice packaged food with vice–virtue bundles, contributing valuable insights to the understanding of consumer behavior and preferences in the context of packaged food choices.

### 6.2. Theoretical Implications

The current research makes several valuable contributions to the existing literature. First, it enhances the current comprehension of mixed bundles. Prior research has primarily focused on the vice–virtue bundles among the served-on-a-plate foods; this research investigates their impact in the context of packaged food. This distinction is vital as calorie perception and product purchase are suspected to differ between these two contexts [5,9]. Meanwhile, packaged foods make up a large proportion of the food market, and they increasingly include vice–virtue bundles. This research expands our understanding of mixed vice–virtue bundles by revealing that the inclusion of such bundles in vice packaged food enhances consumers’ purchase intentions. In addition, given that previous research has mainly explained why vice–virtue bundles stimulate purchase intention from the perceived taste and emotional state perspectives [6,21,52], our work broadens the current knowledge of the mediating mechanisms by identifying a mediator from the perceived healthiness view.

Second, our work contributes to the literature on packaged food consumption. Previous literature has investigated the factors influencing purchase intentions for packaged food from various angles, including package appearances such as color [53], shape [26], and pattern complexity [11], as well as package material features like plastic content [54] and smart degree [55]. Additionally, the claims of included ingredients on the packaging, such as botanical ingredients [30] and flavor-giving components [56], have been explored. Our research enriches this line of literature by identifying an original factor, that is, vice–virtue bundles claimed on the packaging, that positively influences consumers’ willingness to buy vice-packaged food.

Third, this literature complements the application of the regulatory focus theory in the field of packaged food consumption. Previous research on consumer behavior has explored the responses of individuals with different regulatory focuses to marketing messages, with promotion-focused individuals tending to focus more on positive aspects and exhibiting trusting attitudes, while prevention-focused individuals lean towards considering negative aspects and displaying skepticism towards messages [17,39,42]. Our study further verifies these conclusions during the exploration of vice packaged food consumption by revealing that promotion-focused consumers are more likely to perceive vice packaged food with vice–virtue bundles (vs. without vice–virtue bundles) as healthier and thus are more willing to buy it, compared to prevention-focused consumers.

### 6.3. Practical Implications

Our research holds several practical implications for policymakers, consumers, and food marketers. First, our study reveals that vice–virtue bundles significantly increase consumers’ perceived healthiness and purchase intentions, which reminds policymakers to develop and enforce policies to monitor and limit the false claims of vice–virtue bundles in the vice packaged food. This step will help prevent the potential misuse of vice–virtue bundles and protect consumers from making biased judgments and blind consumption. Meanwhile, policymakers can consider implementing measures like setting up alerts or other means to guide consumers toward making informed and rational food choices.

Second, consumers should rationally analyze the vice–virtue bundles’ claims to prevent unintended and unwise purchases due to the overestimation of healthiness and biased food calorie estimates. Reducing this irrational consumption has important implications for advancing healthy eating and long-term well-being.

Third, food marketers can promote consumption and boost short-term profits by adding virtue ingredients to vice packaged food and highlighting the features of vice–virtue bundles on the packaging. In addition, our finding that regulatory focus plays a moderating role suggests that vice packaged food with vice–virtue bundles can be promoted to target consumers who are promotion focus. Marketers can also include more descriptions that activate the promotion focus, such as “promote”, “increase”, “contribute to”, “security”, and “benefit”, in advertisements to awaken consumers’ promotion focus and to stimulate their desire to consume vice packaged food with vice–virtue bundles.

Fourth, despite the inclusion of virtue ingredients, vice packaged food with vice–virtue bundles remains at its core an unhealthy product. Thus, the above-mentioned marketing strategies may cause consumers to self-blame later and decrease their loyalty to the brand. To make the construction of vice–virtue bundles a sustainable marketing strategy, marketers can emphasize the presence of virtue ingredients while reminding consumers that the product is still based on vice food. This helps consumers develop a holistic cognition and make rational trade-offs, ultimately reducing future regret and increasing trust in the product and the brand.

### 6.4. Limitations and Future Directions

The current research has several limitations that provide opportunities for future studies. First, while all the experimental materials are solid foods, it would be highly valuable to test the effect of vice–virtue bundles on the willingness to consume liquid foods with the increasing use of vice–virtue bundles in liquid foods, such as carbonated drinks with an addition of vegetable extract. Furthermore, given that this research only examines individuals’ preferences for vice packaged food that is added with virtue ingredients, the effect of adding vice ingredients inside virtue packaged food (e.g., adding chocolate to whole meal bread and covering the nut with sugar-coated) on consumers’ attitudes deserves to be explored.

Second, while we examined people’s attitudes toward vice packaged food with vice–virtue bundles by communicating the information about vice–virtue bundles to consumers in the form of a claim printed on the packaging (a type of visual cue), follow-up research could emphasize the characteristics of vice–virtue by stimulating the auditory (e.g., introducing the vice–virtue bundles through broadcast announcements) and olfactory (releasing odors of virtue ingredients) [57,58,59].

Third, although our research tested the moderating role of regulatory focus, individual characteristics (e.g., product knowledge, curiosity, gender) still need to be discussed in future research. For instance, it is revealed that product knowledge partly determines individuals’ cognition of products [60], suggesting that it may also play a moderating role in the relationship between vice–virtue bundles and perceived healthiness.

Fourth, some of the food stimuli currently chosen are highly specific to the food culture of the study location, underscoring the need for future food selections to encompass a broader scope of consumers’ comprehensive understanding of foods and diets. For instance, items like carbonated drinks, confectionery, French fries, and fried chicken can serve as vices, while fruits and vegetables are better suited to represent virtues. Moreover, expanding our experiments to encompass a broader spectrum of cultures, such as examining consumers from Western countries, would provide a more compelling demonstration of cultural universality.

Finally, incorporating longitudinal designs and big data analysis into our research is crucial to enhance the reliability of the findings. For example, investigating the quantity of specific food purchased by participants over a one-month period, downloading the sales volume, the duration of time those items were viewed, or the frequency of clicks on particular items from some shopping websites [61]. The inclusion of these methodologies, coupled with qualitative techniques [62], such as instructing participants who bought either vice packaged food without vice–virtue bundles or vice packaged food with vice–virtue bundles to individually document the reasons for their purchases and then extracting essential insights from their explanations, holds the potential to more effectively captures the mediating mechanisms [63].

## 7. Conclusions

Packaged foods have experienced significant growth in market share [64,65], and the concept of vice–virtue bundles has gained traction in vice packaged food as they offer a balance between pleasure and healthiness [21]. Consequently, understanding how consumers respond to vice packaged food with vice–virtue bundles and why becomes crucial to food marketing and regulation. Our four experimental studies demonstrate that consumers exhibit a heightened inclination to purchase vice packaged food with vice–virtue bundles than vice packaged food without vice–virtue bundles. The proposed effect is mediated by the perceived healthiness. However, the amplified intention to purchase vice packaged food with vice–virtue bundles, as well as the mediating effect of perceived healthiness, becomes subdued among consumers adopting a prevention focus on comparison to those with a promotion focus. These findings not only enhance the existing research and theories related to vice–virtue bundles, packaged food consumption, and regulatory focus, but also offer practical insights to guide rational consumption, promote healthier dietary choices, and inform marketing strategies for packaged foods. Nevertheless, the current research does exhibit certain limitations in aspects such as the variety of stimuli employed, the scope of the research target, the manner of bundle presentation, the search for boundary conditions, and the empirical methodology, which highlights areas that require further investigation in future research endeavors.

## Figures and Tables

**Figure 1 foods-12-03270-f001:**
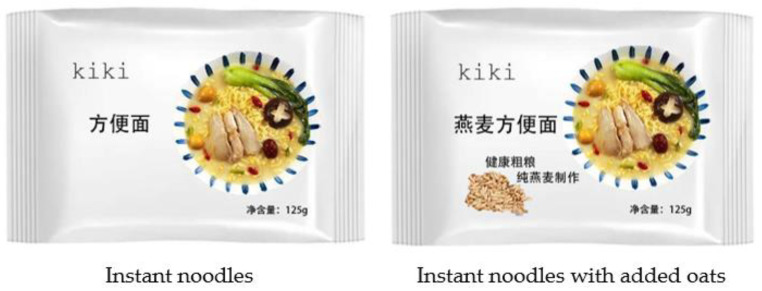
The packaged food used in Experiment 1. Note. In the left picture, the text on the left reads “instant noodles” and the text in the lower right corner reads “net content: 125 g”. In the right picture, the text on the left reads “oat instant noodles”, the text in the lower left corner reads “healthy grains, made of pure oats”, and the text in the lower right corner reads “net content: 125 g”.

**Figure 2 foods-12-03270-f002:**
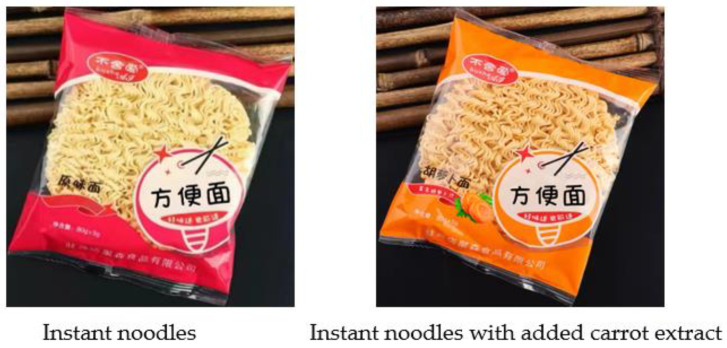
The packaged food used in the follow-up experiment. Note. In the left picture, the text on the left reads “original flavor instant noodles” and “net content: 80 g ± 5 g”, the text at the bottom reads “Zhumadian Jusen Food Company Limited”, and the text on the right reads “instant noodles”. In the right picture, the text on the left reads “carrot instant noodles”, “net content: 80 g ± 5 g”, and “rich in carrot extract”, the text at the bottom reads “Zhumadian Jusen Food Company Limited”, and the text on the right reads “instant noodles”.

**Figure 3 foods-12-03270-f003:**
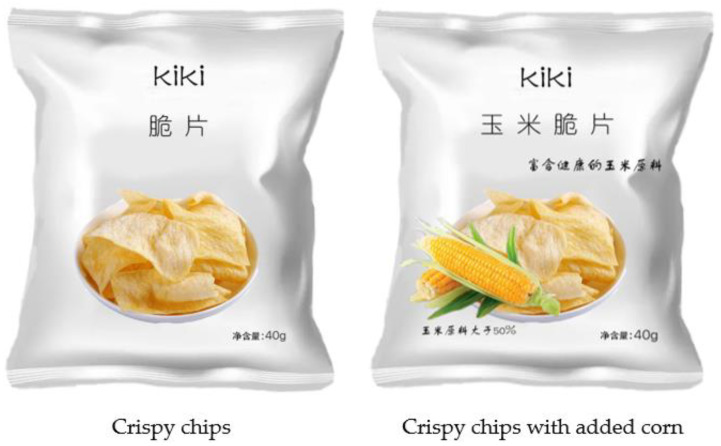
The packaged food used in Experiment 2. Note. In the left picture, the text on the top reads “crispy chips” and the text in the lower right corner reads “net content: 40 g”. In the right picture, the text on the top reads “corn crispy chips” and “rich in healthy corn ingredients”, the text in the lower left corner reads “corn ingredients is more than 50%”, and the text in the lower right corner reads “net content: 40 g”.

**Figure 4 foods-12-03270-f004:**
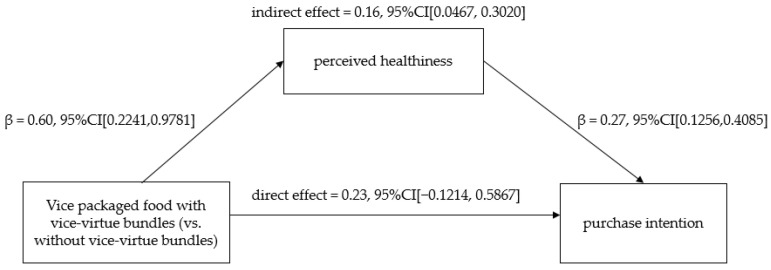
The results of mediation analysis.

**Figure 5 foods-12-03270-f005:**
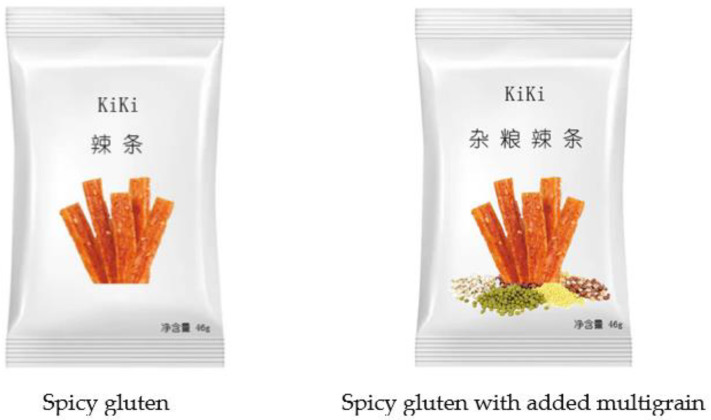
The packaged food used in Experiment 3. Note. In the left picture, the text on the top reads “spicy gluten” and the text in the lower right corner reads “net content: 46 g”. In the right picture, the text on the top reads “multigrain spicy gluten”and the text in the lower right corner reads “net content: 46 g”.

**Figure 6 foods-12-03270-f006:**
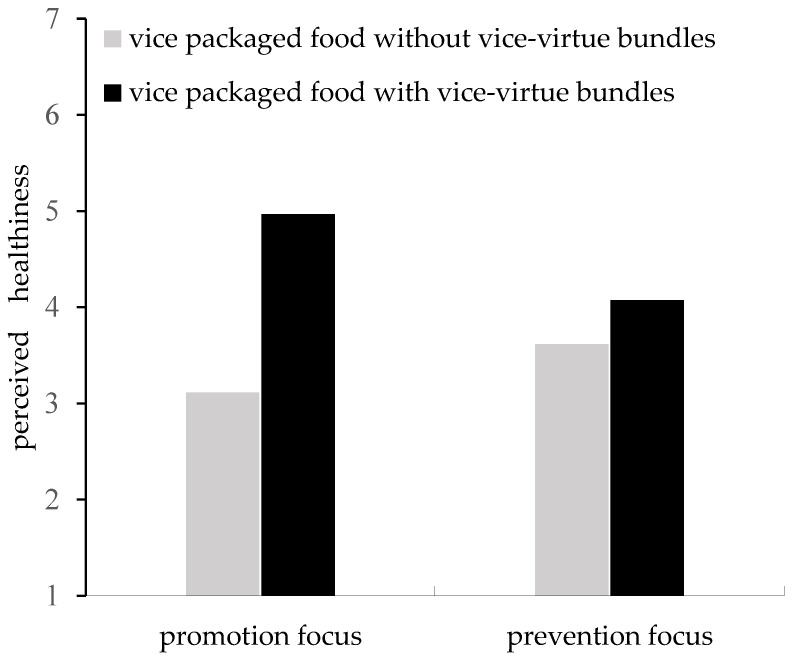
The moderating effect of regulatory focus.

## Data Availability

The data presented in this study are available upon request from the corresponding authors. The data presented in this study are available on request from the corresponding author. The data is not publicly available due to the need to maintain the confidentiality of study participants.

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
