# Peer review of "The Effect of Vice–Virtue Bundles on Consumers’ Purchase Intentions for Vice Packaged Foods: Evidence from Randomized Experiments"

_foods, 2023, doi:10.3390/foods12173270_

Round 1

Reviewer 1 Report

Comments and Suggestions for Authors

Foods 2548823.pper.review.v!.vicious

Overall, there is a literature on vice-virtue concerning healthy pleasurable foods and healthy foods. But I am not familiar with the word vicious used in this context – in the English language vicious is a very strong word meaning fierce. I have never seen vicious used as the opposite of virtue or of health.

Study 1.

Line 148. Method 3.1. I do not like the use of students in such research. Most students do not eat like most working people; they often do not shop and prepare meals. They  do not represent most consumers.

Line 166. “I perceive the food is vice” and “I perceive the food is virtue” – these statements would not make any sense to most consumers, and would certainly not apply to the foods studied.

Line 178. Not a big difference between the means.

Study 2.

Line 212. Not a big difference between the means

Study 3.

Line 249. Did the label say “added miscellaneous grains” – what does that mean?

Line 267. A bigger difference than we have seen in the other studies (M 5.30 and M 4.27).

Discussion.

Line 306. “prefer vicious packaged goods with vice-virtue bundles” – yes that was found, but it is not surprising. And the authors need to address the fact that virtue was always obtained by adding ingredients – maybe consumers rate adding ingredients as a positive virtue.

Also, all of the tested foods were packaged carbohydrates – why? None of them is particularly healthy.

Author Response

Response to Reviewer 1’s comments

Dear Reviewer,

Thank you for providing us with your valuable comments and suggestions on our manuscript (Manuscript ID: foods-2548823). We highly appreciate the time and effort you have dedicated to reviewing our work. Based on your comments, we have made revisions to our manuscript as described below.

Point 1: Overall, there is a literature on vice-virtue concerning healthy pleasurable foods and healthy foods. But I am not familiar with the word vicious used in this context – in the English language vicious is a very strong word meaning fierce. I have never seen vicious used as the opposite of virtue or of health.

Response 1: We would like to express our gratitude to you for pointing out this issue. Generally speaking, "vice" can carry a range of significances and connotations contingent upon the surrounding context. While a prevalent definition of "vice" pertains to immoral or wicked conduct (Baumeister & Exline, 1999), it also encompasses alternative interpretations linked to indulgence (Read et al., 1999; Wertenbroch, 1998).

For our present focus, "vice" is utilized not to denote malevolence, but rather a form of self-gratification or excess in particular pursuits. In this framework, "vice" refers more to a type of self-indulgent behavior, and numerous studies explore "vice" as a facet of indulgence (Chernev & Gal, 2010; Chernev, 2011; Lu et al., 2023; Masters & Mishra, 2019; Mishra & Mishra, 2011; Read et al., 1999; Verma et al., 2016; Wertenbroch, 1998; Yang et al., 2021). It is construed as an indulgence, juxtaposed with virtue, representing the antithesis of these two concepts.

Elaborating on the specifics of these two concepts from the perspective of indulgence, as outlined by Wertenbroch (1998):

Vice: This category encapsulates elements that yield instant gratification but carry the potential for prolonged or postponed harm. Opting for vice often implies self-indulgence, and thus it's aptly translated as indulgence. Vices pertain to actions, choices, or behaviors that yield immediate pleasure, contentment, or satisfaction, but often result in enduring harm or unfavorable outcomes. Engaging in vices may encompass indulging in activities that are pleasurable in the short term but carry negative consequences in the long run. Examples of vices include overindulging in unhealthy food, excessive spending on luxury items, procrastination, and engaging in addictive behaviors.

Virtue: This classification pertains to elements that maximize long-term utility but lack the allure of instantaneous gratification. Choosing virtue typically entails exercising successful self-control. Virtues embody behaviors, actions, or choices that prioritize long-term well-being and self-discipline. Opting for virtues often involves sacrificing immediate gratification in favor of reaping more substantial, enduring benefits. Examples of virtues encompass maintaining a healthy lifestyle, saving money for the future, practicing discipline, and displaying kindness and compassion.

In summary, the contrast between vice and virtue often involves the tension between short-term satisfaction and long-term fulfillment (Read et al., 1999; Wertenbroch, 1998). This might involve one’s physical, mental, or financial well-being (Verma et al., 2016; Wertenbroch, 1998). Within the realm of food, "vice" typically alludes to items that compromise health—namely, unhealthy foods, while “virtue” usually refers to healthy foods (Chernev & Gal, 2010; Haws & Liu, 2016; Lu et al., 2023; Wertenbroch, 1998; Yang et al., 2021).

Furthermore, "vice" and "virtue" frequently emerge in studies related to vice-virtue bundles, describing unhealthy and healthy foods (Carroll et al., 2018; Liu et al., 2015). Therefore, in the current research, we use vice to denote unhealthy food and virtue to denote healthy food. Moreover, we also explain the meanings of vices and virtues to the participants in the experiments.

Baumeister, R.F.; Exline, J.J. Virtue, personality, and social relations: Self-control as the moral muscle. J Pers. 1999, 67(6), 1165-1194. https://doi.org/10.1111/1467-6494.00086

Carroll, K.A.; Samek, A.; Zepeda, L. Food bundling as a health nudge: Investigating consumer fruit and vegetable selection using behavioral economics. Appetite 2018, 121, 237-248. https://doi.org/10.1016/j.appet.2017.11.082

Chernev, A. Semantic anchoring in sequential evaluations of vices and virtues. J. Consum. Res. 2011, 37(5), 761-774. https://doi.org/10.1086/656731

Chernev, A.; Gal, D. Categorization effects in value judgments: Averaging bias in evaluating combinations of vices and virtues. J. Mark. Res. 2010, 47(4), 738-747. https://doi.org/DOI 10.1509/jmkr.47.4.738

Haws, K.L.; Liu, P.J. Combining food type(s) and food quantity choice in a new food choice paradigm based on vice-virtue bundles. Appetite 2016, 103, 441-449. https://doi.org/10.1016/j.appet.2015.11.012

Liu, P.J.; Haws, K.L.; Lamberton, C.; Campbell, T.H.; Fitzsimons, G.J. Vice-Virtue bundles. Manage. Sci. 2015, 61(1), 204-228. https://doi.org/10.1287/mnsc.2014.2053

Lu, F.C.; Park, J.; Nayakankuppam, D. The influence of mindset abstraction on preference for mixed versus extreme approaches to multigoal pursuits. J. Consum. Psycho. 2023, 33(1), 62-76. https://doi.org/10.1002/jcpy.1296

Masters, T.M.; Mishra, A. The influence of hero and villain labels on the perception of vice and virtue products. J. Consum. Psychol. 2019, 29(3), 428-444. https://doi.org/10.1002/jcpy.1085

Mishra, A.; Mishra, H. The influence of price discount versus bonus pack on the preference for virtue and vice foods. J. Mark. Res. 2011, 48(1), 196-206. https://doi.org/10.1509/jmkr.48.1.196

Read, D.; Loewenstein, G.; Kalyanaraman, S. Mixing virtue and vice: Combining the immediacy effect and the diversification heuristic. J. Behav. Decis. Mak. 1999, 12(4), 257-273. https://doi.org/10.1002/(SICI)1099-0771(199912)12:43.3.CO;2-Y

Verma, S.; Guha, A.; Biswas, A. Investigating the pleasures of sin: The contingent role of arousal-seeking disposition in consumers' evaluations of vice and virtue product offerings. Psychol. Mark. 2016, 33(8), 620-628. https://doi.org/10.1002/mar.20904

Wertenbroch, K. Consumption self-control by rationing purchase quantities of virtue and vice. Mark. Sci. 1998, 17(4), 317-337. https://doi.org/10.1287/mksc.17.4.317

Yang, S.G.; Xu, Q.; Jin, L.Y. Sweet or sweat, which should come first: How consumption sequences of vices and virtues influence enjoyment. Int. J. Res. Mark. 2021, 38(4), 1073-1087. https://doi.org/10.1016/j.ijresmar.2021.03.002

Study 1.

Point 2: Line 148. Method 3.1. I do not like the use of students in such research. Most students do not eat like most working people; they often do not shop and prepare meals. They do not represent most consumers.

Response 2: We would like to express our gratitude to you for pointing out this issue and to apologize for our thoughtlessness.

We conducted a follow-up experiment (see details in Supplementary Material and brief description in Section 3 of the revised manuscript), which used employees of a company as participants, real packaged foods as stimuli, and actual behavioral responses as the measure. This experiment, along with Experiment 2, raises more robust evidence for a main effect.

Since this new experiment was conducted last, significantly later than Experiments 1, 2, and 3, we made it a follow-up experiment followed by Experiment 1 while the previous Experiment 1 was retained. Moreover, even though Experiment 1 was carried out with participants from a college student population, the stimuli employed (in the form of a meticulously controlled self-designed packaging) and the measurement methods (utilizing scales) diverged from those utilized in this follow-up experiment. This discrepancy enables the two experiments to synergize and offer more robust evidence in support of the hypothesis.

In addition, the purpose of Experiment 1 was to test Hypothesis 1. As stated in the results section of Experiments 2 and 3, both Experiments 2 and 3, in addition to testing either Hypothesis 2 or Hypothesis 3, also both repetitively tested Hypothesis 1 (i.e., the conclusions reached in Experiment 1 were tested repetitively). Notably, the participants in Experiments 2 and 3 were not students, but rather samples from social platforms and samples from several communities. Thus, Experiments 2 and 3 further provided more reliable evidence for Hypothesis 1 by examining samples from a wider range of sources.

Point 3: Line 166. “I perceive the food is vice” and “I perceive the food is virtue” – these statements would not make any sense to most consumers, and would certainly not apply to the foods studied.

Response 3: We would like to express our gratitude to you for pointing out this issue and apologize for our non-detailed statement in the materials and procedures section. During the pretest, we provided participants with explanations regarding the contextual meanings of virtues and vices. This clarification was offered prior to their formal response to the question, that is, in the section of the highlighting response requirements. The statement of this step was incorporated into Section 3.1.2 of the revised manuscript.

Point 4: Line 178. Not a big difference between the means.

Response 4: We would like to express our gratitude to you for pointing out this issue. We have conducted a follow-up experiment, which uses real foods as stimuli, employs actual behavioral responses as the measure, and is conducted among company staff. This new experiment has also verified Hypothesis 1. Therefore, this research has examined three hypotheses through several experiments incorporating distinct stimuli and measures, and involving samples from diverse sources. These experiments differed in the measurement stimuli used, the measures of purchase intention, and the source of the sample.

In addition, we also state in the limitations of the discussion section that future research can enrich the research methodology “incorporating longitudinal designs and big data analysis into our research is crucial to enhance the reliability of the findings. For example, investigating the quantity of specific food purchased by participants over a one-month period, or downloading the sales volume, the duration of time those items were viewed, or the frequency of clicks on particular items from some shopping websites. The inclusion of these methodologies, coupled with qualitative techniques, such as instructing participants who bought either pure vicious packaged food or vicious packaged food with vice-virtue bundles to individually document the reasons for their purchases and then extracting essential insights from their explanations, holds the potential to more effectively captures the mediating mechanisms. Furthermore, expanding our experiments to encompass a broader spectrum of cultures, such as examining consumers from western countries, would provide a more compelling demonstration of cultural universality”.

Study 2.

Point 5: Line 212. Not a big difference between the means (Study 2).

Point 6: Line 267. A bigger difference than we have seen in the other studies (M 5.30 and M 4.27). (Study 3)

Response 5&6: We would like to appreciate you for bringing this matter to our attention.

In our manuscript, during the transcription of results from the SPSS output, an unfortunate error occurred. Specifically, we mistakenly transcribed the mean and standard deviation of the control group's purchase intention from Experiment 2 into the results of Experiment 3 in the manuscript, but the mean and standard deviation of Experiment 2 were not transcribed incorrectly as well as other statistics such as the standard error, F-value, and p-value. We deeply apologize for this oversight.

We have taken corrective measures in the revised version of the manuscript, rectifying the results pertaining to Experiment 3 (“participants in the experimental condition (M = 5.30, SD = 1.28) reported greater purchase intention than those in the control condition (M = 4.72, SD = 1.5 8; F (1, 247) = 9.92, p = 0.002; η2 = 0.04”). To ensure the accuracy of our findings, several authors conducted comprehensive reviews of the complete paper and ascertained that no further errors persisted. We apologize for this error and humbly ask that you may forgive us for it.

In addition, while the findings that the purchase intention in the experimental group and control groups was significantly different in all studies implied that the hypothesis regarding the main effect (i.e., H1: Consumers are more willing to purchase vicious packaged food with vice-virtue bundles than pure vicious packaged food) was confirmed in all studies, your inquiry concerning the difference in means guides us to a deeper thinking, an aspect that holds significant value for us. After correcting the incorrect transcription, we found that the difference between the control and experimental groups’ purchase intention in Experiment 3 was slightly smaller (Mexperimental = 5.30, Mcontrol = 4.72) than that in Experiment 1 (Mexperimental = 4.71, Mcontrol = 4.06) and Experiment 2 (Mexperimental = 5.01, Mcontrol = 4.27). We think that the small divergence in means among the various studies may stem from disparities in the sample size or the stimuli employed (i.e., the presentation of pure vicious packaged food and the added virtuous ingredients). It is also possible that there are additional factors that contribute to the magnitude of the differences in means across studies.

Lastly, to enhance the generality and robustness of our conclusions to abbreviate the degree of interference due to uncontrollable factors, we added a follow-up experiment. This new experiment used real packaged food (pure instant noodles and instant noodles with added carrot extract) as stimuli and used company’s staff as participants. Encouragingly, these results further substantiated our hypothesis.

In general, we tried to improve the reliability and robustness of our findings, especially of the main effects (i.e., the effect of the independent variable on the dependent variable), by conducting multiple experiments using distinct stimuli and measures, and involving sample from diverse sources.

Study 3.

Point 7: Line 249. Did the label say “added miscellaneous grains” – what does that mean?

Response 7: We would like to express our gratitude to you for pointing out this issue and to apologize for our unclear description.

We aimed to test the effect of vicious packaged food with vice-virtue bundles (i.e., the vicious packaged food with added virtuous ingredients) on purchase intentions compared to pure vicious packaged food (the vicious packaged food without special additions). Thus, in our experiment, participants in the experimental group saw the spicy gluten package presenting the claim “spicy gluten with added miscellaneous grains”, while participants in the control group saw the spicy gluten package with only the claim of spicy gluten. (as in Figure 4)

Miscellaneous grains are also known as coarse cereals, miscellaneous grain crops, side crops, or food grains other than wheat and rice, which belongs to relatively healthy food. Miscellaneous grains refer to grain and bean crops other than the five major crops: rice, wheat, corn, soybeans, and potatoes. It mainly contains: sorghum, grain, buckwheat, oats, mung beans, red beans, broad beans, peas, black beans, and so on. They are characterized by a short growing period, small planting area, special planting areas, lower yields, and are generally rich in nutrients, and are usually considered healthy food. The participants of this experiment were from China, and most Chinese people are more aware of mixed grains. In addition, our pre-test results also showed that participants do consider miscellaneous grains as a kind of virtuous food.

In addition, although spicy glutens (a food with heavy oil and spicy) are usually made from flour derived from wheat, other flours derived from other grains can also be used as raw materials for crafting spicy gluten. This piece of knowledge is widespread, especially in China. Therefore, Chinese participants can easily realize that spicy glutens contain miscellaneous grains when they saw a label on the package that reads “miscellaneous grain spicy gluten” or “ spicy gluten with added miscellaneous grains”.

Discussion.

Point 8: Line 306. “prefer vicious packaged goods with vice-virtue bundles” – yes that was found, but it is not surprising. And the authors need to address the fact that virtue was always obtained by adding ingredients – maybe consumers rate adding ingredients as a positive virtue.

Response 8: We would like to express our gratitude to you for pointing out this issue. As stated in the introduction section, the vicious packaged food with vice-virtue bundles refers to the vicious packaged food with an addition of virtuous ingredients, that is, incorporating vicious ingredients into pure virtuous packaged food. Indeed, in this research, the vice-virtue bundles refers to the bundling of vicious packaged food with virtuous ingredients.

In vicious packaged foods, when virtue ingredients are added to the original pure vicious packaged product, a bundles of vice and virtue is generated.

Such a food will present both vice and virtue in its packaging, and people shall not see and perceive virtue or vice separately in isolation, but will perceive vice and virtue as a whole, i.e., vice-virtue bundles, so the evaluation of this packaged food will not come from vice or virtue alone, but from bundles as a whole (e.g., oat instant noodles, carrot instant noodles, corn crispy chip, miscellaneous-grains spicy gluten). Thus, our research also just puts a lot of emphasis on the impact of bundles claims as a whole versus pure vicious claims on consumers.

This related description was also further emphasized in the discussion section of the revised manuscript.

Point 9: Also, all of the tested foods were packaged carbohydrates – why? None of them is particularly healthy.

Response 9: We would like to express our gratitude to you for pointing out this issue and to apologize for our unclear explanation.

As described in lines 40 through 44, vice-virtue bundles occur in two contexts, one is incorporating vicious ingredients into pure virtuous packaged food (i.e., virtuous packaged food with vice-virtue bundles), and the other is adding virtuous ingredients to pure vicious packaged food (i.e., vicious packaged food with vice-virtue bundles). In addition, as stated in lines 54 through 57, our research focuses on vicious packaged food with vice-virtue bundles. Therefore, the basis of the packaged food using in our measurement is unhealthy food, in other words, the packaged food used in our measurement is the food that has healthy ingredients added to unhealthy food, such as spicy gluten with an addition of miscellaneous grains, instant noodles with an addition of oat.

Carbohydrates such as crisps, spicy gluten, and instant noodles are more likely to be perceived as unhealthy. Pure vicious packaged food and vicious packaged food with added virtuous ingredients are also more common in the broader category of carbohydrates. Therefore, the selection of such foods as stimuli is a better fit for our research object. Therefore, the basis of the tested foods in this research were all packaged carbohydrates. In addition, the foods we chose to use for the measurements also passed the pretest before the formal experiments. The relevant explanations were further emphasized in Section 3.1.2 of the revised manuscript as well.

Finally, we also mentioned in the discussion section that we will add liquid-based foods for measurement in the future to further improve the generality of our findings.

Thanks for your valuable comments and suggestions. I am very excited and grateful that you provide us with a lot of valuable comments and an opportunity to revise. Moreover, we really learned a lot of knowledge and the rigorous scientific research spirit from your comments and suggestions, for which we sincerely express our deep gratitude.

We hope these explanations and revisions can alleviate your concerns. In addition, we fully agree and allow you to make any revisions to the manuscript. Thanks again for your assistance and your support.

Reviewer 2 Report

Comments and Suggestions for Authors

TITLE AND ABSTRACT

-       The title of the manuscript is neither concise nor relevant. It should be better written to make it easier for readers to understand.

-       The title does not identify whether the study reports trial data (human or animal), or is a systematic review, meta-analysis or replication study. It does indicate the type of intervention (The impact of vice-virtue bundles on purchase intention for vicious packaged food).

-       In this case, the title of the clinical trial does not identify whether it is a randomized or non-randomized trial.

-       The title does not include the study population (university students). However, it would be interesting to know the exact age range to which the study is directed.

-       The abstract is well structured as it is written in a single paragraph and does not present headings, but the subsections of the design, methods, results, and conclusions of the trial are indicated. However, the type of design used during the experimental studies, as well as the methods used, could be further specified.

-       The abstract should have a total of about 200 words maximum and in this case, it is well within this word limit (150 words).

-       In the abstract the results shown are present and substantiated in the main text.

KEY WORDS

-       The keywords are correct; they have to be three to ten relevant keywords after the abstract. In this study, 7 keywords specific to the article and common within the subject discipline are observed.

INTRODUCTION

-       The introduction shows the importance of the study since many people when choosing packaged products find themselves in a complicated situation to manage where they do not know whether to opt for healthy but not tasty foods or for unhealthy but tasty foods, therefore, not knowing whether to prioritize health or the immediate craving they have at the time of purchase.

-       The purpose of the work and its importance has to be defined, including the specific hypotheses being tested. A section on Theories and hypotheses has been created, where the hypotheses are indicated, however, following the Foods magazine regulations, in the introduction section they must be indicated briefly, as well as if there are any controversial hypotheses, they must also be indicated.

-       The conclusions of the work are mentioned, but the main objective of the work is not mentioned briefly.

-       Most of the references used in this section are not recent, but they do follow the theme of the study. It is recommended to use the most recent references possible.

-       In general, the introduction is quite incomplete, although it shows a good overview of the current state of the subject, but more emphasis should be placed on the structure indicated in the journal's guidelines, as well as indicating the purpose, hypotheses and objectives of the work.

-       The Theories and hypotheses section could be merged somehow with the Introduction section so that all the necessary information would be in a single section with the title Results.

RESULTS

-       This section does not exist. According to the regulations of the journal, it is necessary that one of the main sections of the paper be exclusively the results section.

-       In this section we would strongly recommend a flow chart, for each group, the number of university students that were randomly assigned and included in the main analysis.

-       Indicate in the same flow chart mentioned above for each group, the losses and exclusions after randomization for each hypothesis, together with the reasons.

-       The results of the study are shown individually in 3 subsections (3.2,4.2,5.2) within the 3 types of experimental studies that have been carried out. Within these 3 subsections of results, the dates that define the recruitment periods of the university students are not indicated, nor are the dates when the appropriate tests were carried out for the study.

-       In addition, it would be advisable to indicate the results for each group, the estimated effect size and its precision as 95% confidence intervals. It is preferable to express the results as a 95% CI rather than as a p value, since these intervals give a truer idea of the magnitude of the differences observed and their clinical significance.

-       It would be advisable to include in this Results section a representation of the initial demographic and clinical characteristics of each participating group in the 3 studies performed. This representation is usually a table that corresponds to Table 1, in accordance with the standard observed in most Randomized Clinical Trials (RCT).

DISCUSSION

-       Throughout the theoretical implication’s subsection, the contributions that the study has had according to the authors to the existing literature are indicated; however, it would be advisable to make a comparison with current relevant studies related to the exposed subject matter. In addition to mentioning them throughout the discussion section, in order to confirm these contributions that are affirmed.

-       It is indicated in section 6.3 that the study presents a series of limitations that can serve to provide opportunities for future studies on this subject. For example, all the experimental studies in the present work have been carried out with solid foods, so it would also be convenient to draw conclusions from more studies with liquid foods.

-       As mentioned above, the authors do indicate the limitations presented in a subsection exclusively for these, but do not indicate whether they have used any method to minimize any of these limitations. However, it would be advisable to indicate possible solutions also for the rest of the limitations indicated in this section.

-       It would be advisable to make a final interpretation consistent with the results, with a balance of benefits and harms, and considering other relevant evidence.

-       This section can be combined with Results.

MATERIALS AND METHODS

-       This section does not exist; according to Foods magazine regulations, it is necessary that only the Materials and methods section appears as the main section.

-       The methods of the study are shown individually in 3 subsections (3.1,4.1,5.1) within the 3 types of experimental studies that have been carried out. Within these 3 methods subsections, the dates defining the recruitment periods of the university students and the dates when the appropriate tests were performed for the study are not indicated.

-       This methods subsection does not indicate the type of study it is, as well as the mechanism used to implement the random assignment sequence, describing the steps performed to hide the sequence until the interventions were assigned.

-       In the three methods subsections, the participants, i.e., the sample size for each experimental study, are noted. In the first study 172 participants, in the second study 169 participants and in the third study 249 participants. But the selection and exclusion criteria of the participants are not indicated.

-       It does not indicate who generated the randomization sequence, who selected the participants and who assigned the participants to the interventions.

-       In Experiments 1 and 2, the items of the scales used in each experiment are explicitly shown; however, for Experiment 3, in which a scale of 10 items and 7 points was used, these items are not indicated. Therefore, it is recommended that they be indicated as in the two previous experiments.

-       There is no section on materials.

-       It is recommended that the materials used during the study be indicated and that these be shown in detail throughout the section, thus allowing others to replicate it in exactly the same way.

-       It is recommended to create a single section under the heading of materials and methods, where all the necessary information is indicated, following the Foods journal guidelines.

CONCLUSION

-       Conclusion (optional): not included. The researchers have not made a brief conclusion in this section. It would be advisable to do so due to the complexity of the discussion section.

REFERENCES:

-       The structure of the references used in the journal articles used by the authors is correct. The format indicated by the journal for this type of references is followed.

-       Example of the guide of the journal of the study, to introduce the references correctly in the references section of the work:

o   Journal Articles: Author 1, A.B.; Author 2, C.D. Title of the article. Abbreviated Journal Name Year, Volume, page range.

-       In the second reference introduced throughout the work, specifically it is observed in the introduction section, a "," would be missing to separate reference number 2 from reference number 3, since it is currently in the following form: [23,4].

-       Another aspect to be taken into account and which is perfectly complied with throughout all the references is that the name of the journal in the reference should appear in abbreviated form.

Author Response

Response to Reviewer 2’s comments
Dear Reviewer,
Thank you for providing us with your valuable comments and suggestions on our manuscript (Manuscript ID: foods-2548823). We highly appreciate the time and effort you have dedicated to reviewing our work. Based on your comments, we have made revisions to our manuscript as described below.

TITLE AND ABSTRACT
Point 1: The title of the manuscript is neither concise nor relevant. It should be better written to make it easier for readers to understand.
The title does not identify whether the study reports trial data (human or animal), or is a systematic review, meta-analysis or replication study. It does indicate the type of intervention (The impact of vice-virtue bundles on purchase intention for vicious packaged food).
In this case, the title of the clinical trial does not identify whether it is a randomized or non-randomized trial.
The title does not include the study population (university students). However, it would be interesting to know the exact age range to which the study is directed.
Response 1: We would like to appreciate you for bringing this matter to our attention. We have changed the title from "Balancing hedonic and healthiness: The impact of vice-virtue bundles on purchase intention for vicious packaged food" to "The effect of vice-virtue bundles on consumers’ purchase intentions for vicious packaged foods: Evidence from randomized experiments". 
Specifically, we have added "consumers'" to indicate that it is a research of studying people, added "experiments" to indicate that it is reporting experimental data, and added "randomized" to indicate that the experiments in this paper are randomized experiments. Moreover, we have tried to improve the conciseness of the title while still including the necessary information by deleting "Balancing hedonic and healthiness" 
While Experiment 1 utilized a sample of students, the samples in both Experiment 2 and Experiment 3 were not confined to students alone. In addition, none of the samples for the follow-up experiments added in this revision were students. Consequently, the present research did not exclusively center around students, thereby warranting the exclusion of "students" from the title.
Lastly, the samples employed across our four experiments exhibited diversity, encompassing a broad spectrum of ages. This diversity made it difficult to establish a uniform age range and thus can not show an age range in the title

Point 2: The abstract is well structured as it is written in a single paragraph and does not present headings, but the subsections of the design, methods, results, and conclusions of the trial are indicated. However, the type of design used during the experimental studies, as well as the methods used, could be further specified. The abstract should have a total of about 200 words maximum and in this case, it is well within this word limit (150 words). In the abstract the results shown are present and substantiated in the main text.
Response 2: We would like to appreciate you for bringing this matter to our attention. We sincerely apologize for our previous incomplete statement. We have further specified the type of design used during the experimental studies and the methods used in the revised abstract within the word limit (200 words).

KEY WORDS
Point 3: The keywords are correct; they have to be three to ten relevant keywords after the abstract. In this study, 7 keywords specific to the article and common within the subject discipline are observed.
Response 3: We would like to express our gratitude to your recognition.

INTRODUCTION
Point 4: The introduction shows the importance of the study since many people when choosing packaged products find themselves in a complicated situation to manage where they do not know whether to opt for healthy but not tasty foods or for unhealthy but tasty foods, therefore, not knowing whether to prioritize health or the immediate craving they have at the time of purchase. The purpose of the work and its importance has to be defined, including the specific hypotheses being tested. A section on Theories and hypotheses has been created, where the hypotheses are indicated, however, following the Foods magazine regulations, in the introduction section they must be indicated briefly, as well as if there are any controversial hypotheses, they must also be indicated.
Point 7: In general, the introduction is quite incomplete, although it shows a good overview of the current state of the subject, but more emphasis should be placed on the structure indicated in the journal's guidelines, as well as indicating the purpose, hypotheses and objectives of the work.
Response 4&7: We would like to express our gratitude to you for pointing out this issue. We have further highlighted and clarified the background of the research at the theoretical and practical levels and the objectives of the research in the first and second paragraphs. Moreover, we have further expounded upon the research hypotheses in the third paragraph to provide a more comprehensive understanding.

Point 5: The conclusions of the work are mentioned, but the main objective of the work is not mentioned briefly.
Response 5: We would like to appreciate you for bringing this matter to our attention. We have added a conclusion section (Section 7) that states the background and object of the research, and summarizes the findings, contributions, and limitations of the current research in the revised manuscript. In addition, we have added a summary of findings section, which further summarizes the results of all experiments, compares and corroborates them with previous studies, and the efforts of enhancing robustness and generality.

Point 6: Most of the references used in this section are not recent, but they do follow the theme of the study. It is recommended to use the most recent references possible.
Response 6: We would like to appreciate you for bringing this matter to our attention. The previous version contained 24 (45.28%) references published between 2018 and 2023. we have deleted some of the references published prior to 2018 and added some references published between 2018 and 2023, now featuring 35 (53.84%) references published between 2018 and 2023. Notably, the six references published between 1990 and 2000 was not deleted, since they primarily provide crucial references to some of the concepts, theories, and scales in this research.

Point 8: The Theories and hypotheses section could be merged somehow with the Introduction section so that all the necessary information would be in a single section with the title Results.
Response 8: We would like to appreciate you for pointing out this issue. We've observed that certain articles, particularly those with concise introductions, do not feature distinct subheadings like "Introduction" and "Theory and Hypotheses." Instead, they directly present the theory and rationale behind the hypotheses shortly after introducing the introductory content. 
However, our introduction was relatively comprehensive and long, which presented the research question and object through the lens of real phenomena and existing research gaps, offered a brief overview of the research hypotheses, and culminated by summarizing the practical and theoretical implications, with the intention of engaging the reader's interest and facilitating a comprehensive understanding of the research in a concise manner. Moreover, the revised introduction was further refined in terms of the objectives and hypotheses. Therefore, the introduction section and the hypotheses and theories section were given separate headings in the current paper. 
In future research endeavors, we will strive to condense the introduction while amalgamating the theory and hypotheses sections into a single cohesive unit.

RESULTS
Point 9: This section does not exist. According to the regulations of the journal, it is necessary that one of the main sections of the paper be exclusively the results section.
Response 9: We would like to appreciate you for bringing this matter to our attention and apologize for this omission. 
In accordance with the journal’s regulations and with reference to recent papers published in this journal that share our research paradigm (multiple behavioral experiments) (Liang et al., 2022; Meersseman et al., 2021), we have added a summary of findings section in the general discussion section, which summarizes the results of the three main experiments and the one follow-up experiment and compares and corroborates them with previous studies.
In addition, in our previous manuscript, each sub-experiment included a results section describing the detailed findings of that sub-experiment.
Liang, S.; Qin, L.; Zhang, M.; Chu, Y.; Teng, L.; He, L. Win Big with Small: The Influence of Organic Food Packaging Size on Purchase Intention. Foods 2022, 11, 2494. https://doi.org/10.3390/foods11162494 
Meersseman, E.; Geuens, M.; Vermeir, I. Take a Bite! The Effect of Bitten Food in Pictures on Product Attitudes, Purchase Intentions, and Willingness to Pay. Foods 2021, 10, 2096. https://doi.org/10.3390/foods10092096

Point 10: In this section we would strongly recommend a flow chart, for each group, the number of university students that were randomly assigned and included in the main analysis. 
Point 11: Indicate in the same flow chart mentioned above for each group, the losses and exclusions after randomization for each hypothesis, together with the reasons.
Response 10 &11: We would like to appreciate you for pointing out this issue. With reference to recent papers published in Foods journal and other papers regarding consumer behavior that share our research paradigm, we have added four flow charts, showing the number of participants that were randomly assigned in the experiment, the number of participants included in the main analysis, the materials and procedures. These flow charts are large and therefore placed in Supplementary Material (Figures 1 to 4).
Specifically, the following criteria were used to recruit subjects to participate in the experiment and to select subjects for formal analysis
Prior to conducting the experiment, individuals with psychiatric diseases, any deadly diseases, or blindness were excluded from participation, as these specific categories were ineligible for involvement in the study. However, the number of those who were excluded during the recruitment process was not counted.
Following the experiment's completion, subjects who incorrectly answered the attention test questions were subsequently excluded. These participants were deemed ineligible for inclusion in the formal analysis. The number of participants excluded in this step was labeled in the flow chart of Supplementary Material and participants section in the revised manuscript.

Point 12: The results of the study are shown individually in 3 subsections (3.2,4.2,5.2) within the 3 types of experimental studies that have been carried out. Within these 3 subsections of results, the dates that define the recruitment periods of the university students are not indicated, nor are the dates when the appropriate tests were carried out for the study.
Response 12: We would like to appreciate you for bringing this matter to our attention and to apologize for this omission. We have added the periods of recruiting participants and the testing date in Experiments 1 and 2, and the follow-up experiment. 
We have also added the testing date in Experiment 2. Experiment 2 collected data by distributing a link to the questionnaire on a social media platform, so the timing of the test was synchronized with the recruitment of participants. 

Point 13: In addition, it would be advisable to indicate the results for each group, the estimated effect size and its precision as 95% confidence intervals. It is preferable to express the results as a 95% CI rather than as a p value, since these intervals give a truer idea of the magnitude of the differences observed and their clinical significance.
Response 13: We would like to appreciate you for bringing this matter to our attention and we extremely agree with your viewpoint. We have added 95% confidence intervals in the results section of each study in the revised manuscript. However, we performed the ANOVA using SPSS, which does not generate confidence intervals but provides effect sizes, and it results in a large number of published papers analyzed by this method would not report confidence intervals. As a result, except for the ANOVA outcomes, we included confidence intervals for the findings of the mediation and moderation analyses.

Point 14: It would be advisable to include in this Results section a representation of the initial demographic and clinical characteristics of each participating group in the 3 studies performed. This representation is usually a table that corresponds to Table 1, in accordance with the standard observed in most Randomized Clinical Trials (RCT).
Response 14: We would like to appreciate you for bringing this matter to our attention. In behavioral experiments related to exploring consumer behavior, it is common to collect information on people's age, gender, educational background, level of consumer spending, etc. Our experiment also collects information on these demographics.
We enriched each experiment with the gender and age details of all participants, as well as the gender and age breakdown within each group in the revised manuscript. 
Furthermore, we compiled a table that presented a more comprehensive overview of the four experiments’ demographic information, encompassing gender, age, educational background, and monthly consumption that were collected during the experiment. However, we included this table (Table 1) in Supplementary Material since it is very large. Meanwhile, we stated that "see detailed demographics in Supplementary Material" in Section 3.1 of the revised manuscript.

DISCUSSION
Point 15: Throughout the theoretical implication’s subsection, the contributions that the study has had according to the authors to the existing literature are indicated; however, it would be advisable to make a comparison with current relevant studies related to the exposed subject matter. In addition to mentioning them throughout the discussion section, in order to confirm these contributions that are affirmed.
Response 15: We would like to appreciate you for pointing out the issue. We have added a summary of findings section in the general discussion section. In this section, we made a comparison with current relevant studies related to the exposed subject matter and used previous research or theory to further confirm our findings, in which the points included are largely consistent with those in the theoretical implications section. Specifically, we further summarized our results. Then, we compared and validated the results from Experiments 1 to 3 and the follow-up experiment with prior investigations on the influence of virtue-virtue bundles on consumption and the impact of on-package claims on consumer attitudes. Next, we contrasted the outcomes of Experiments 2 to 3 with findings from earlier studies related to the “health halo” phenomenon generated by virtue claims and the perception of health-enhancing consumption, highlighting their parallels. Furthermore, we aligned the conclusions of Experiment 3 with regulatory orientation theory, elucidating their coherence. 

Point 16: It is indicated in section 6.3 that the study presents a series of limitations that can serve to provide opportunities for future studies on this subject. For example, all the experimental studies in the present work have been carried out with solid foods, so it would also be convenient to draw conclusions from more studies with liquid foods. As mentioned above, the authors do indicate the limitations presented in a subsection exclusively for these, but do not indicate whether they have used any method to minimize any of these limitations. However, it would be advisable to indicate possible solutions also for the rest of the limitations indicated in this section.
Response 16: We would like to appreciate you for pointing out the issue and to apologize for the previous incomplete description. We have further expanded on the limitations and provided illustrative examples to demonstrate potential solutions in new Section 6.4 (i.e. previous Section 6.3).

Point 17: It would be advisable to make a final interpretation consistent with the results, with a balance of benefits and harms, and considering other relevant evidence. This section can be combined with Results.
Response 17: We would like to appreciate you for pointing out the issue. In each sub-experiment, a results section include to describe the detailed findings of that sub-experiment. 
In addition, we have added a summary of results section in the general discussion section, which summarizes the results of the three main experiments and the one follow-up experiment and compares and corroborates them with previous studies. 
Specifically, we further summarized our results. Then, we compared and validated the results from Experiments 1 to 3 with prior investigations on the influence of virtue-virtue bundles on consumption and the impact of on-package claims on consumer attitudes. Next, we contrasted the outcomes of Experiments 2 to 3 with findings from earlier studies related to the “health halo” phenomenon generated by virtue claims and the perception of health-enhancing consumption, highlighting their parallels. Furthermore, we aligned the conclusions of Experiment 3 with regulatory focus theory, elucidating their coherence. Finally, we also highlighted our efforts to enhance the robustness of the discovery by utilizing a range of experimental tools. 

MATERIALS AND METHODS
Point 18: This section does not exist; according to Foods magazine regulations, it is necessary that only the Materials and methods section appears as the main section.
Response 18: We would like to appreciate you for bringing this matter to our attention. Since our research has several different sub-experiments and each experiment has its own specific methods and materials. Therefore, we presented methods and materials in each experiment. However, in the previous manuscript, this section was included in the methods section of each experiment. 
In the revised manuscript, we divided the methods section into two sections, "Participants" and " Materials and procedures" and presented them under headings. In addition, we further supplemented the content in the materials and procedures section, clarifying the order of every step, and presenting the stimuli and full items of the scales used.

Point 19: The methods of the study are shown individually in 3 subsections (3.1,4.1,5.1) within the 3 types of experimental studies that have been carried out. Within these 3 methods subsections, the dates defining the recruitment periods of the university students and the dates when the appropriate tests were performed for the study are not indicated.
Response 19: We would like to appreciate you for bringing this matter to our attention and to apologize for this omission. We have added the periods of recruiting university students and the testing date in Experiments 1 and 2, and the follow-up experiment. We have also added the testing date in Experiment 2. Experiment 2 collected data by distributing a link to the questionnaire on a social media platform, so the timing of the test was synchronized with the recruitment of participants.

Point 20: This methods subsection does not indicate the type of study it is, as well as the mechanism used to implement the random assignment sequence, describing the steps performed to hide the sequence until the interventions were assigned.
Point 21: It does not indicate who generated the randomization sequence, who selected the participants and who assigned the participants to the interventions.
Response 20 & 21: We would like to appreciate you for bringing this matter to our attention. We have indicated the type of study in all method sections. 
In addition, the recruitment of participants and selection of participants were both done by authors who had participated in this research, which is also stated in the revised manuscript.
Finally, the random assignment sequences are executed by the primary tester or automatically generated by the data collection platform. Specifically, in Experiment 1, Experiment 3, and the follow-up experiment, a research assistant who did not know the objective of this research served as the primary tester for this study. He instructed participants to select a slip at random from a set of ten, each labeled with numbers from 1 to 10. Their assignment to either the experimental or control condition depended on whether the drawn number was odd or even. Notably, participants were not informed about their condition allocation throughout the experiment. In Experiment 2, the random sequence was generated automatically by this data collection platform and refrained from disclosing participants' allocation to either the experimental or control group. The description regarding who generated the randomization sequence, the mechanism used to implement the random assignment sequence, and the steps performed to hide the sequence until the interventions were assigned were added in the method section of each experiment.

Point 22: In the three methods subsections, the participants, i.e., the sample size for each experimental study, are noted. In the first study 172 participants, in the second study 169 participants and in the third study 249 participants. But the selection and exclusion criteria of the participants are not indicated.
Response 22: We would like to appreciate you for bringing this matter to our attention. We recruited participants who did not suffer from psychiatric diseases or any deadly diseases and who had normal vision, that is, people who showed psychiatric diseases or any deadly diseases, and blind people could not participate in our experiments. Subsequently, participants who fulfilled the experiment but did not successfully clear the attention checks were excluded. For example, if a question required selecting the fifth option and a participant failed to do so, they would not pass the attention assessment and, consequently, be removed from the study. We have further explained the requirements for recruiting participants and screening subjects in the “participants” section of the revised manuscript.

Point 23: In Experiments 1 and 2, the items of the scales used in each experiment are explicitly shown; however, for Experiment 3, in which a scale of 10 items and 7 points was used, these items are not indicated. Therefore, it is recommended that they be indicated as in the two previous experiments.
Response 23: We would like to appreciate you for bringing this matter to our attention and apologize for this omission. In Section 5.1.1 of the revised manuscript, we showed these items.

Point 24: There is no section on materials. It is recommended that the materials used during the study be indicated and that these be shown in detail throughout the section, thus allowing others to replicate it in exactly the same way. It is recommended to create a single section under the heading of materials and methods, where all the necessary information is indicated, following the Foods journal guidelines.
Response 24: We would like to appreciate you for bringing this matter to our attention. In the revised manuscript, we divided the methods section into two sections, “Participants” and “Materials and procedures” and presented them under headings. In addition, we further supplemented the content in the materials and procedures section, further clarifying the order of every step, and presenting the stimuli and full items of the scales used.

CONCLUSION
Point 25: Conclusion (optional): not included. The researchers have not made a brief conclusion in this section. It would be advisable to do so due to the complexity of the discussion section.
Response 25: We would like to appreciate you for bringing this matter to our attention. We have added a conclusion that states the background and object of the research, and summarizes the findings, contributions, and limitations of the current research in the revised manuscript. 

Point 26: REFERENCES
The structure of the references used in the journal articles used by the authors is correct. The format indicated by the journal for this type of references is followed.
Example of the guide of the journal of the study, to introduce the references correctly in the references section of the work:
Journal Articles: Author 1, A.B.; Author 2, C.D. Title of the article. Abbreviated Journal Name Year, Volume, page range.
In the second reference introduced throughout the work, specifically it is observed in the introduction section, a "," would be missing to separate reference number 2 from reference number 3, since it is currently in the following form: [23,4].
Another aspect to be taken into account and which is perfectly complied with throughout all the references is that the name of the journal in the reference should appear in abbreviated form.
Response 26: We would like to express our gratitude to your recognition and to apologize for our omission regarding punctuation. We have corrected this formatting error.

Thanks for your valuable comments and suggestions. I am very excited and grateful that you provide us with a lot of valuable comments and an opportunity to revise. Moreover, we really learned a lot of knowledge and the rigorous scientific research spirit from your comments and suggestions, for which we sincerely express our deep gratitude. 
We hope these explanations and revisions can alleviate your concerns. In addition, we fully agree and allow you to make any revisions to the manuscript. Thanks again for your assistance and your support.

Round 2

Reviewer 1 Report

Comments and Suggestions for Authors

Thank you for the revision. The paper is much improved and reads more clearly. I still have concern about the word vicious - I checked your reference (6) and they do not appear to use that term for vice-virtue. I recommend that you introduce your argument for using the word vicious in the Introduction, and again in the final Discussion at the end of the paper. English speaking readers will need this assistance in understanding your use of the word vicious.

Comments on the Quality of English Language

minor editing needed

Author Response

Response: We would like to express our gratitude to you for pointing out this issue and apologize for our misunderstandings.

We had thought that “vice” and “vicious” were similar words, except that the latter is more of an adjective but their meanings are almost identical. But now, I realize that this cognition is wrong. As stated in your previous comments, in the English language vicious is a very strong word meaning fierce.

It is precisely because we have interpreted the meanings of “vice” and “vicious” to be the same that we have misinterpreted the meaning of you. We previously thought that you did not understand why vice or vicious was used to describe unhealthy, so we explained the relationship between vice and unhealthy through a large body of literature. However, we now realize that you agree with the use of vice to represent unhealthy food and simply believes that vicious is not appropriate to represent unhealthy food.

Indeed, as stated in you this comment, a great deal of previous literature on vice-virtue bundles uses vice rather than vicious, e.g., vice option, vice offerings, vice product, vice components. In addition, when describing healthy foods, they also directly use "virtue" instead of "virtuous", e.g. virtue option, virtue offerings, virtue product, virtue components (Liu et al., 2015; Verma et al., 2016; Yang et al., 2021). Thus, we have changed “vicious” to “vice” and changed “virtuous” to “virtue”.

Thanks again for bringing this matter to our attention and your continued assistance and support. Thank you very much.